# Baxter Permutation Process

**Masahiro Nakano    Akisato Kimura    Takeshi Yamada    Naonori Ueda**
NTT Communication Science Laboratories, NTT Corporation
`{masahiro.nakano.pr,akisato.kimura.xn,takeshi.yamada.bc,naonori.ueda.fr}`
`@hco.ntt.co.jp`

## Abstract

In this paper, a Bayesian nonparametric (BNP) model for Baxter permutations (BPs), termed BP process (BPP) is proposed and applied to relational data analysis. The BPs are a well-studied class of permutations, and it has been demonstrated that there is one-to-one correspondence between BPs and several interesting objects including floorplan partitioning (FP), which constitutes a subset of rectangular partitioning (RP). Accordingly, the BPP can be used as an FP model. We combine the BPP with a multi-dimensional extension of the stick-breaking process called the *block-breaking process* to fill the gap between FP and RP, and obtain a stochastic process on arbitrary RPs. Compared with conventional BNP models for arbitrary RPs, the proposed model is simpler and has a high affinity with Bayesian inference.

## 1   Introduction

Bayesian nonparametric (BNP) methods can overcome the model complexity problem of machine learning tasks, as they can be regarded as an analysis of finite subsets of potentially infinite data using infinite-dimensional probabilistic models, i.e., stochastic processes. Indeed, a variety of stochastic processes have been proposed and applied to various real-world tasks. However, in general, it is not easy to define and control new BNP models, because they should satisfy certain stringent conditions,[1] such as projectivity [10, 43, 44, 45, 16], exchangeability [6, 7, 31, 32], and conditional projectivity [44, 45]. In this paper, we develop a BNP model of Baxter permutations (BPs). This model involves new stochastic processes and is applied to relational data analysis.

Currently, there are a variety of BNP models for relational data analysis. Recent excellent surveys can be found in [20, 46]. Conventional models are broadly classified into three categories: (a) clustering through rectangular partitioning (RP), (b) factor analysis (extraction of multiple clusters) [14, 47, 52, 40, 30, 13], and (c) analysis using more flexible structures [5, 21, 24, 37, 38, 26, 19, 23, 22]. This paper focuses on the first category. Its advantage is that all clusters are disjoint rectangles characterized by products of subsets of each dimension of the relational data, which can be easily interpreted. For RP models, the infinite relational model (IRM) [33] and the Mondrian process (MP) [49, 48] have been widely studied and applied to real world applications. However, these models cannot represent arbitrary RPs. That is, their supports are limited to some subsets of all possible RPs (Figure 1, second and third). In contrast, the Gilbert tessellation (GT) [27, 39] and the rectangular tiling process (RTP) [42] have been proposed for arbitrary RPs with no restrictions (Figure 1, fourth). However, for the GT, it is known that the statistical behavior of it is notoriously difficult to analyze [12]. For the RTP, it constructs a probabilistic generative model that directly generates a RP of grids with infinite size. However, it has too complicated procedures for the model construction due to its projectivity property, and is not well-suited for Bayesian inference.

**Contributions -** The aim of this paper is to construct a new BNP model for arbitrary RPs, so that it has a simple description and high affinity with Bayesian inference. We first discuss RPs and

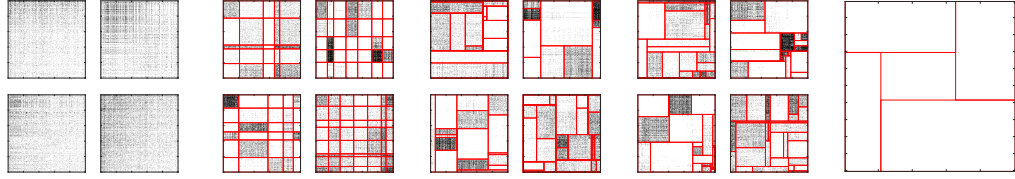

Figure 1: Relational data and three classes of rectangular partitioning discussed in combinatorics [41]. (From left to right) **First:** Samples of (binary) relational data. **Second: Regular grid -** The rows and columns are partitioned into clusters. Each block is characterized by the product of the row and column clusters. **Third: Hierarchical -** Partitionings are expressed as binary trees where nodes represent a vertical or horizontal separation of a rectangle into two disjoint rectangles. **Fourth: Arbitrary -** No restrictions are required. This class is obtained by the proposed method. **Fifth:** Example not included in either hierarchical or regular grid.

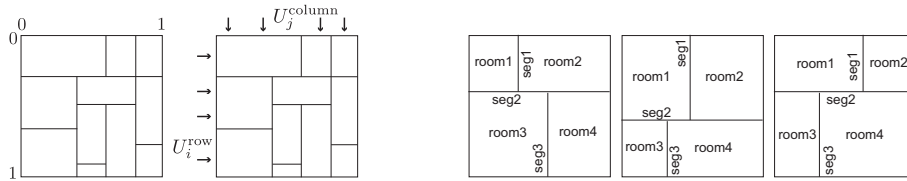

Figure 2: **Left:** Illustration of Aldous-Hoover-Kallenberg representation of exchangeable array. **Right:** Illustration of definition of FP. These different RP samples are equivalent in the sense of FP.

floorplan partitioning (FPs) (plainly, FPs constitute a subset of RPs). Surprisingly, there is one-to-one correspondence between FPs and BPs [18], which are a class of permutations [9]. Based on this fact, the main contributions of this paper are to propose new stochastic processes shown as follows:

- The BP process (BPP): We construct a generative probabilistic BP model, the *projectivity* property of which ensures the existence of its limit, that is, an infinite BP model. By the one-to-one correspondence between BPs and FPs, the BPP can also be used as an FP model.

- The block-breaking process (BBP): We combine the BPP with *block-breaking process*, a multi-dimensional extension of the stick-breaking process [51], to fill the gap between FP and RP. We apply the BBP to the Aldous-Hoover-Kallenberg representation [6, 29, 32] to obtain a BNP model for arbitrary RPs of relational data.

## 2 Preliminaries

### 2.1 Relational models, Rectangular partitioning (RP), and Floorplan partitioning (FP)

In this paper, RP can be regarded as partitions of $[0, 1] \times [0, 1]$ such that all blocks form disjoint rectangle clusters of $[0, 1] \times [0, 1]$. By the Aldous-Hoover-Kallenberg (AHK) representation theorem [6, 29, 32] for *exchangeable* arrays, the RP has high affinity with the BNP model. Figure 2 (left) shows an illustration of the AHK representation. We assume that an observation of relational data consists of rows indexed by $\{1, \ldots, N\}$ and columns indexed by $\{1, \ldots, M\}$. Given some BNP models for RP, a generative probabilistic model of the relational data can be easily constructed as follows. First, we draw an RP sample based on some BNP models. Then we draw independent and identically distributed (i.i.d.) uniform random variables:

$$U_i^{\text{row}} \sim \text{Uniform}([0, 1]) \ (i = 1, 2, \ldots, N), \quad U_j^{\text{column}} \sim \text{Uniform}([0, 1]) \ (j = 1, 2, \ldots, M). \quad (1)$$

Finally, the cluster assignment of each element, with row and column indexed by $i$ and $j$, respectively, is specified by the block on $[0, 1] \times [0, 1]$ to which the point $(U_i^{\text{column}}, U_j^{\text{column}})$ belongs. According to the AHK representation, we can focus on constructing BNP models for RP.

In addition, we introduce another important concept, namely FP. In an FP, the size of each rectangle block of the room partition is irrelevant. We follow the definition in [50] regarding the notion of equivalence for two FP samples. Figure 2 (right) shows an example. Given an FP sample $f$, a *segment*

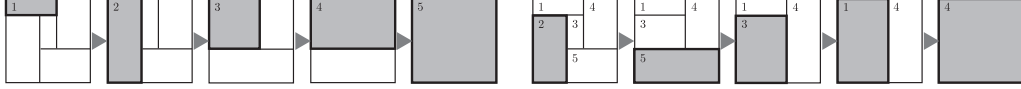

Figure 3: Illustration of Algorithm 1. **Left**: The *top-left* room is labeled as 1, and deleted by the *top-left* room deletion operator. Likewise, the top-left room is labeled as 2, . . . , and delete it hereinafter. As a result, all rooms are labeled by 1, 2, . . . . **Right**: The BP is obtained by repeatedly extracting the label of the *bottom-left* room and deleting it using the *bottom-left* room deletion.

(cut) $s$ supports a *room* (block) $r$ in $f$ if $s$ contains one of the edges of $r$. We say that $s$ and $r$ have a top-, left-, right-, or bottom-seg-room relation if $s$ supports $r$ from the respective direction. Two FP samples are equivalent if there is a labeling of their rooms and segments such that they hold the same seg-room relations under the labeling. Thus, three FP samples in Figure 2 (right) are equivalent.

## 2.2 Baxter permutations

In 1964, Glen Baxter introduced a class of permutations in the context of fixed points for the composition of commuting functions, which now bear his name [9]. A *Baxter permutation* (BP) on $\{1, 2, \ldots, n\}$ ($n \in \mathbb{N}$) is a permutation $\pi = (\sigma_1 \sigma_2 \ldots \sigma_n)$ for which there are no quadruples of indices $i < j < j + 1 < k$ such that

$$\sigma_j < \sigma_k < \sigma_i < \sigma_{j+1} \quad \text{or} \quad \sigma_{j+1} < \sigma_i < \sigma_k < \sigma_j. \tag{2}$$

For example, a permutation $\pi = (\sigma_1 \sigma_2 \ldots \sigma_8) = \mathbf{61832547}$ is not *Baxter*, since it contains a quadruple $1 < 3 < 4 < 8$ such that $\sigma_4 = \mathbf{3} < \sigma_1 = \mathbf{6} < \sigma_8 = \mathbf{7} < \sigma_3 = \mathbf{8}$. For more intuitions, consider the case of $n = 4$. All permutations of $\{1, 2, 3, 4\}$ are listed as follows:

$$1234, 1243, 1324, 1342, 1423, 1432, 2134, 2143, 2314, 2341, \mathbf{2413}, 2431,$$
$$3124, \mathbf{3142}, 3214, 3241, 3412, 3421, 4123, 4132, 4213, 4231, 4312, 4321. \tag{3}$$

A BP avoids the patterns, **3142** and **2413**. Such patterns with prescribed adjacencies are often termed *vincular* patterns.

The BPs are a well-studied class of permutations, which have a number of nice properties associated to them. We briefly review the most relevant two properties of the BPs in this paper. First, there is a one-to-one correspondence between BPs and several combinatorial objects, such as twin binary trees, plane bipolar orientations and some type of three non-intersecting paths on a grid [18, 25]. Especially, in this paper, we focus on its application to the FP. We show a direct bijection between FP and BP, introduced by [55, 50]. Second, we introduce some useful properties related to the enumeration of the BPs, and describe the enumeration algorithm proposed in [15].

### 2.2.1 Mapping from floorplan partitioning to Baxter permutation

We first define the following operator on FP. Given an FP sample with $n$ rooms in $[0, 1] \times [0, 1]$ as its bounding rectangle, we can obtain a FP sample with $(n - 1)$ rooms by using the following *room deletion* operator, introduced by [28]. The **top-left** room deletion is defined as follows:

**Definition** *(Top-left room deletion).* Let $f$ be an FP sample with $n > 1$ rooms and let $r$ be the top-left room in $f$. (1) If the bottom-right corner of $r$ has a "⊣" junction, then we delete $r$ from $f$ by shifting the bottom edge upwards while keeping all "⊤" junctions on the bottom edge attached, until the edge reaches the bounding rectangle. (2) If the bottom-right corner of $r$ has a "⊥" junction, then we delete $r$ from $f$ by shifting the right edge leftwards while keeping all "⊢" junctions on the right edge attached, until the edge reaches the bounding rectangle.

Similarly, we can define the **bottom-left** room deletion operator. Then, according to the top-left and bottom-left room deletion operators, we can obtain the mapping from the FP into the BP.

Figure 3 shows an illustration of Algorithm 1. The output of Algorithm 1 is always a BP, as shown in [17] (Lemma 3.6). Moreover, the mapping corresponding to Algorithm 1 is injective [17] (Lemma 3.7). Next we move on to the mapping from the BP to the FP.

---
**Algorithm 1** MAPPING FLOORPLAN PARTITIONING TO BAXTER PERMUTATION
---

**Input**: Floorplan partitioning $f$ with $n$ rooms.

· Assign labels $1, 2, \ldots, n$ in ascending order into $n$ rooms by repeatedly labeling the *top-left* room and applying *top-left* room deletion operator to it (Figure 3, left).

**Output**: Return the permutation of labels obtained by repeatedly extracting the label of the *bottom-left* room and applying the *bottom-left* room deletion operator into it (Figure 3, right).

---

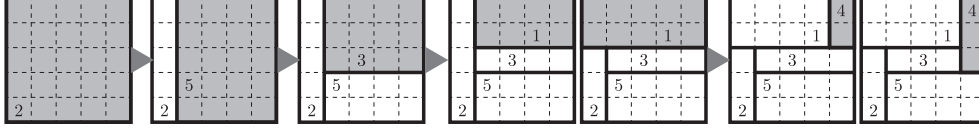

Figure 4: Illustration of Algorithm 2. A BP sample $\pi = (\sigma_1 \sigma_2 \ldots \sigma_n) = \mathbf{25314}$ is transformed to a FP sample. First, we draw a block labeled as $\sigma_1 = \mathbf{2}$, and construct a $5 \times 5$ grid. Second, since we have $\sigma_2 = \mathbf{5} > \sigma_1 = \mathbf{2}$, we bisect the top-right block by a *vertical* segment at the second grid. Third, since we have $\sigma_3 = \mathbf{3} < \sigma_2 = \mathbf{5}$, we bisect the top-right block by a *horizontal* segment at the third grid. Fourth, we bisect the top-right block by a horizontal segment, and then extend the block $\sigma_4 = 1$ *leftward* at the expense of $\sigma_1 = \mathbf{2}$, since the block $\sigma_1 = \mathbf{2}$ to the left of $\sigma_4 = 1$ has a label greater than $\sigma_i$. Finally, Algorithm 2 obtains the corresponding FP sample to $\mathbf{25314}$.

### 2.2.2 Mapping from Baxter permutation to floorplan partitioning

Given a BP on $\{1, \ldots, n\}$, Algorithm 2 constructs a FP sample with $n$ rooms [17]. As is shown in Figure 4, the algorithm iteratively inserts rooms one by one into the top-right corner of the FP. The $i$-th room is generated by bisecting the previous room, and is labeled according to the $i$-th element in the BP. If the $(i-1)$-th element is smaller (resp., greater) than the current element, the room is bisected vertically (resp., horizontally). The resulting horizontal (resp., vertical) segment is extended leftward (resp., downward) if the room to the left (resp., below) has a greater (resp., smaller) label than that of the current room.

---
**Algorithm 2** MAPPING BAXTER PERMUTATION TO FLOORPLAN PARTITIONING
---

**Input**: Baxter permutation $\pi = (\sigma_1 \sigma_2 \ldots \sigma_n)$.

· Draw a block and label it as $\sigma_1$.

· Construct an $n \times n$ grid within the block.

**for** i=2 to $n$ **do**

    **if** $\sigma_i < \sigma_{i-1}$ **then**

        · Bisect the top-right block by a *horizontal* segment at the $i$-th grid.

        · Label the new top-right block as $\sigma_i$.

        **while** t **do**

            he block $\sigma'$ to the left of $\sigma_i$ has a label greater than $\sigma_i$,· Extend the block $\sigma_i$ *leftward* at the expense of $\sigma'$.

        **end while**

    **else**

        · Bisect the top-right block by a *vertical* segment at the $i$-th grid.

        · Label the new top-right block as $\sigma_i$.

        **while** t **do**

            he block $\sigma'$ below $\sigma_i$ has a label smaller than $\sigma_i$,· Extend the block $\sigma_i$ *downward* at the expense of $\sigma'$.

        **end while**

    **end if**

**end for**

**Output**: Floorplan partitioning with $n$ blocks.

---

### 2.2.3 Enumeration of Baxter permutations

In order to construct a generative BP model, the enumeration algorithm proposed in [15] is quite useful. Here, we briefly review the enumeration process for BPs.

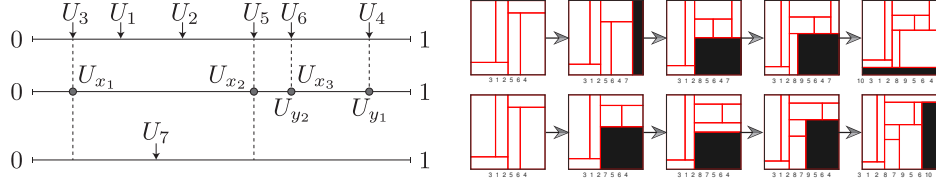

Figure 5: **Left:** Illustration of BPP. Consider a BP $\mathbf{312564} \in \mathcal{Z}_6$ and its latent parameters $U_1, \ldots, U_6$. This BP has left-to-right maxima $x_1 = \mathbf{3} < x_2 = \mathbf{5} < x_3 = \mathbf{6}$ and right-to-left maxima $\mathbf{6} = y_2 > \mathbf{4} = y_1$. If $U_7$ is drawn from the interval $[U_3, U_5]$, then $\mathbf{7}$ is inserted to the immediate left of $\mathbf{5}$ of $\mathbf{312564}$, and the resulting BP on $\{1, \ldots, 7\}$ is $\mathbf{3127564}$. We emphasize that the BP is not equivalent to the order of $U_1, \ldots, U_7$. **Right:** Illustration of FP evolution according to underlying BPP. Two FP samples are growing according to the BPP. Instead of direct transformations from a FP with $n$ blocks to a FP with $n+1$ blocks, the evolution of a FP is obtained only through the underlying evolution of a BP by using Algorithm 2. For example, we consider an evolution of a BP from $\mathbf{312564}$ to $\mathbf{3127564}$. We apply Algorithm 2 to both $\mathbf{312564}$ and $\mathbf{3127564}$, and obtain the corresponding FPs to $\mathbf{312564}$ and $\mathbf{3127564}$, respectively.

The first property is that BPs are closed under removing the largest label, leading to the projectivity property of the BPP for Kolmogorov's extenstion theorem (discussed later in Section 3, Proposition 3.2). We note that this is not immediately obvious, as BPs are given by a vincular pattern, that involves adjacency issues. However, the following was positively proved in [15] ([17], Lemma 3.1):

**Proposition 2.1** *If $\pi = (\sigma_1 \sigma_2 \ldots \sigma_n)$ is a BP on $\{1, \ldots, n\}$, and we remove its largest label $\sigma_i = n$, then the result is also a BP.*

The second issue is a method for generating a BP on $\{1, \ldots, n\}$ from a BP on $\{1, \ldots, n-1\}$. Proposition 2.1 means that every BP on $\{1, \ldots, n\}$ arises from a BP on $\{1, \ldots, n-1\}$ by inserting $n$ into an admissible position. Fortunately such admissible positions were explicitly determined in [15]:

**Proposition 2.2** *Given a BP on $\{1, \ldots, n-1\}$, we consider the BP on $\{1, \ldots, n\}$ by inserting $n$. The admissible positions where $n$ can be inserted are limited to each of the immediate left of the left-to-right[2] maxima, and to each of the immediate right of the right-to-left maxima.*

The third property is whether we can enumerate all possible BPs by the procedure shown in Proposition 2.2, which specifies the support of the BPP (discussed later in Section 3, Proposition 3.1):

**Corollary 2.3** *Consider the generating tree for BP that every node on the $n$-th level corresponds to a BP on $\{1, \ldots, n\}$, and has the children nodes obtained by inserting $(n+1)$ into all admissible positions of the corresponding BP of the parent node, described in Proposition 2.2. For any $n \in \mathbb{N}$, the set of the BPs corresponding to the nodes on the $n$-th level of this generating tree is equivalent to all BPs on $\{1, \ldots, n\}$.*

# 3 Baxter permutation process (BPP)

The first contribution of this study is a BNP model for BPs. Let $\mathcal{Z}_n$ be the set of all BPs on $\{1, \ldots, n\}$. The BPP is a discrete-time Markov process on BPs and generates an object that, on the $n$-th time, corresponds to a BP sample on $\mathcal{Z}_n$. We present an illustrative example of the proposed model. Given the BP sample $\mathbf{312564} \in \mathcal{Z}_6$, we consider the possible BPs obtained by inserting $\mathbf{7}$ into admissible positions. According to Proposition 2.2, these positions are immediately left of the left-to-right maxima $\mathbf{3}, \mathbf{5}, \mathbf{6}$ and immediately right of the right-to-left maxima $\mathbf{4}, \mathbf{6}$, that is,

$$\underbrace{\quad}\mathbf{3\,1\,2}\underbrace{\quad}\mathbf{5}\underbrace{\quad}\mathbf{6}\underbrace{\quad}\mathbf{4}\underbrace{\quad}. \tag{4}$$

As shown in this example, the evolution of the BPP depends on the left-to-right and the right-to-left maxima, as well as the choice of the admissible positions. For notational convenience, we use

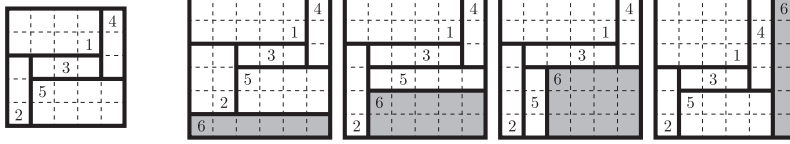

Figure 6: Evolution of FP according to BPP. The left FP corresponds to **25314**. The right four patterns are all possible FPs corresponding to the BPs in $\mathcal{Z}_6$ whose projection onto $\mathcal{Z}_5$ is **25314**. We note again that we do not have direct transformations from the FP corresponding to **25314** to the FPs with 6 block. We apply Algorithm 2 to **625314**, **265314**, **256314** and **253146** independently to obtain the corresponding FPs.

$x_1, x_2, \ldots, x_i$ and $y_1, y_2, \ldots, y_j$ to indicate the left-to-right maxima and the right-to-left maxima of a BP, respectively. In order to describe the evolution of the BPP, we introduce auxiliary variables, consisting of a sequence of independent and identically distributed (i.i.d.) uniform random variables $U_1, U_2, \ldots$ on $[0, 1]$. The resulting BPP sample on the $n$-th time is obtained from $U_1, \ldots, U_n$. Figure 5 provides an illustration. In the following, we will provide a more precise description.

**Model description -** The BPP is a discrete-time Markov process $\pi := (\pi(t_n),\ n \in \mathbb{N})$ over time $t_1, t_2, \ldots$ where each $\pi(t_n)$ is a BP sample on $\mathcal{Z}_n$. The BPP $\pi(t_n)$ on $t_n$ has a collection of latent parameters, consisting of i.i.d. uniform random variables $U_1, \ldots, U_n$ on $[0, 1]$. Given a sample $\pi(t_n) = (\sigma_1 \sigma_2 \ldots \sigma_n)$ generated from $U_1, \ldots, U_n$, a sample $\pi(t_{n+1})$ is drawn as follows. Without loss of generality, we can assume that $\pi(t_n)$ has left-to-right maxima $x_1 < \cdots < x_i = n$ and right-to-left maxima $n = y_j > \cdots > y_1$. We additionally assume that $U_1, \ldots, U_n$ satisfies

$$U_{x_1} < U_{x_2} < \cdots < U_{x_i} = U_n = U_{y_j} < U_{y_{j-1}} < \cdots < U_{y_1}. \tag{5}$$

We note that this assumption is not obvious, and therefore it will be proved by mathematical induction. For convenience, we let $U_{x_0} = 0$ and $U_{y_0} = 1$. The above inequality implies that the real line $[0, 1]$ is divided into intervals $[U_{x_0}, U_{x_1}], [U_{x_1}, U_{x_2}], \ldots [U_{x_{i-1}}, U_{x_i}], [U_{y_j}, U_{y_{j-1}}], \ldots, [U_{y_1}, U_{y_0}]$. Then, the latent parameter $U_{n+1}$ is independently drawn from the uniform distribution on $[0, 1]$. If $U_{n+1}$ is located on the interval $[U_{x_{k-1}}, U_{x_k}]$ $(k = 1, \ldots, i)$, then $(n + 1)$ is inserted to the immediate left of $x_k$. If $U_{n+1}$ is located on the interval $[U_{y_l}, U_{y_{l-1}}]$ $(l = 1, \ldots, j)$, then $(n + 1)$ is inserted to the immediate right of $y_l$. By construction, Equation (5) also holds for $U_1, \ldots, U_{n+1}$. Therefore, by induction, Equation (5) holds for all $n \in \mathbb{N}$.

For example, we consider the BP $\pi(t_6) = \mathbf{312564} \in \mathcal{Z}_6$, as shown in Figure 5. We assume that $U_1 \ldots, U_6$ is drawn as the top of Figure 5 (left). This BP has left-to-right maxima $x_1 = \mathbf{3} < x_2 = \mathbf{5} < x_3 = \mathbf{6}$ and right-to-left maxima $\mathbf{6} = y_2 > \mathbf{4} = y_1$, as shown in the middle. If $U_7$ is drawn on the interval $[U_3, U_5]$, then $\mathbf{7}$ is inserted to the immediate left of $\mathbf{5}$ of $\mathbf{312564}$, and the resulting BP $\pi(t_7) \in \mathcal{Z}_7$ corresponds to $\mathbf{3127564}$. We note that the BP is not equivalent to the order of $U_1, \ldots, U_7$.

**Properties -** The BPP $\pi(t_n)$ can define the probability measures $\mu_n$ on $(\mathcal{Z}_n, \mathbf{2}^{\mathcal{Z}_n})$. In the following, we study some properties of $\mu_n$. All proofs are provided in the supplementary material. First, we study the support of $\mu_n$. It has positive probabilities for any possible BPs.

**Theorem 3.1 (Support).** *For any $n \in \mathbb{N}$ and $z_n \in \mathcal{Z}_n$, we have $\mu_n(z_n) > 0$.*

Subsequently, we prove that by Kolmogorov's extension theorem, the projective limit $\mu_\infty$ of probability measures $\mu_n$ $(n \to \infty)$ exists:

**Theorem 3.2 (Projectivity).** *Let $\langle \mu_n \rangle_{n \in \mathbb{N}}$ be the family of probability measures, derived from the BPP. The projector $Q_{m,n} : \mathcal{Z}_m \to \mathcal{Z}_n$ $(n < m \in \mathbb{N})$ is defined as follows: For a BP on $\{1, \ldots, m\}$, the projector $Q_{m,n}$ removes the largest $(m - n)$ labels of the permutation and generates a new BP on $\{1, \ldots, n\}$. Then, for any $n < m \in \mathbb{N}$ and $A_n \in \mathbf{2}^{\mathcal{Z}_n}$, we have the projectivity[3] property: $\mu_n(A_n) = \mu_m(Q_{m,n}^{-1} A_n)$. Accordingly, by Kolmogorov's extension theorem, the family of probability measures $\langle \mu_n, Q_{m,n} \rangle_{n \le m \in \mathbb{N}}$ is uniquely extended to the projective limit probability measure $\mu_\infty$ of the BP on $\{1, 2, \ldots\}$.*

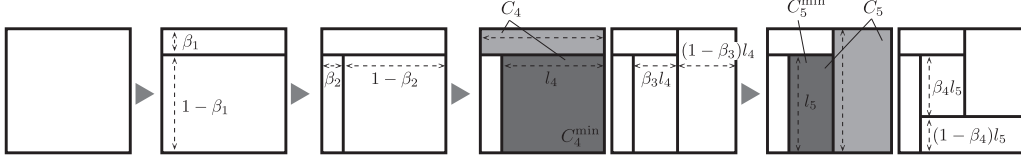

Figure 7: **Left**: Illustration of BBP. The BBP sequentially adds a new bottom-right block into the current rectangular partitioning. For visibility, $C_2, C_2^{\min}, C_3, C_3^{\min}$ are omitted.

## 4 Block-breaking process (BBP)

The BPP can also be used for FP, according to Algorithm 2. However, we have to fill the gap between FP and RP to construct a BNP model based on the AHK theorem. Our strategy is to introduce size adjusting parameters to generate sized blocks of RP from corresponding size-less rooms of FP, which are generated by the BPP. As shown in Figure 6, the evolution of the BPP corresponds to adding a new bottom-right room to the FP. We additionally introduce a sequence of i.i.d beta random variables into the BPP to control the size of the rooms of the FP drawn from the BPP. As in the "stick-breaking process (SBP) of $[0,1]$" [51], the new process is termed *block-breaking process* (BBP) of $[0,1] \times [0,1]$.

**High-level sketch -** The BBP can be broadly interpreted as a multi-dimensional extension of the SBP. We recall that the SBP generates infinite number of sticks of a line $[0,1]$ by recursively drawing a beta random variable $\beta$ and breaking the remaining stick at a ratio of $\beta : (1-\beta)$. Plainly, the BBP replaces the line $[0,1]$ and the sticks of the SBP with the bounding rectangle $[0,1] \times [0,1]$ and rectangle blocks, respectively. The central difficulty of the construction of the BBP unlike the SBP is to have to additionally care about to which directions a new partition should be added recursively. Therefore, we employ the BPP to navigate the evolution of the underlying FP. Following this intuition, we now provide a more precise description.

**Model description -** The BBP is a discrete-time Markov process $b := (b(t_n), \ n \in \mathbb{N})$ over time $t_1, t_2, \ldots$ where each $b(t_n)$ is an RP sample with $n$ blocks. The BBP $b(t_n)$ on $t_n$ has a collection of latent parameters, consisting of i.i.d. uniform random variables $U_1, \ldots, U_n$ on $[0,1]$, and i.i.d. beta random variables $\beta_1, \ldots, \beta_{n-1}$. Figure 7 shows an illustration of the generative BBP model. We consider an RP sample $r(t_{n-1})$ obtained from $U_1, \ldots, U_{n-1}$ and $\beta_1, \ldots, \beta_{n-2}$, and an FP sample $f(t_{n-1})$ with $(n-1)$ rooms, also obtained from $U_1, \ldots, U_{n-1}$ according to the BPP. Given $b(t_{n-1})$ and $f(t_{n-1})$, a sample $b(t_n)$ at the next time $t_n$ is drawn as follows. We first draw $U_n$ and $\beta_{n-1}$ from the uniform and the beta distributions, respectively. If the right-bottom corner of the $(n-1)$-th room of $f(t_n)$ is on the **left** (or **top**) side of the right-bottom corner of the $n$-th room of $f(t_n)$, then let $C_n$ be the set of all blocks (light gray and dark gray in Figure 7) of $b(t_n)$ such that the corresponding rooms of $f(t_n)$ are adjacent to the **left** (or **top**) of the $n$-th room of $f(t_n)$ . Let $C_n^{\min}$ be a block (dark gray in Figure 7) in $C_n$ with the minimum **width** (or **height**) $l_n$. The $n$-th block of the RP is generated by cutting blocks in $C_n$ so that the $n$-th block has a **width** (or **height**) $(1 - \beta_{n-1})l_n$.

**Properties -** As is well known, the SBP-based mixture model (for sequence partitioning) has the following two useful properties. (a) It can express arbitrary partitions of any finite observations. (b) For sufficiently small $\epsilon > 0$, the infinitely many sticks on $[1-\epsilon, 1]$ do not contribute the finite observation data, and the *active* partitions are concentrated on $[0, 1-\epsilon]$. These properties are carried over into the BBP $b = (b(t_1), b(t_2), \ldots)$. By construction, the top-left corner locations of all blocks of $b(t_n)$ are invariant on $t \geq t_n$. This leads to the two useful properties of the BPP-based relational model which is obtained by applying the limit $b(t_\infty)$ to the intermediate random function on $[0,1] \times [0,1]$ of the AHK representation (described in Section 2.1). (a) One is the support of the BPP. The BBP covers arbitrary RPs: this can be easily deduced from the aforementioned property of the BBP constructively. (b) The other is concerning the number of *active* blocks of $b(t_\infty)$ for finite observations. We consider a finite observation matrix consisting of rows indexed by $\{1, \ldots, N\}$ and columns indexed by $\{1, \ldots, M\}$. Let $U_{\max}^{\text{row}}$ and $U_{\max}^{\text{column}}$ be $\max\{U_i^{\text{row}} \mid i = 1, \ldots, N\}$ and $\max\{U_j^{\text{column}} \mid j = 1, \ldots, M\}$, respectively. By construction of the BBP, there exists a natural number $k < \infty$ such that the top-left corner of the $k$-th block of $b(t_\infty)$ is located in the region $[U_{\max}^{\text{row}}, 1] \times [U_{\max}^{\text{column}}, 1]$ with probability 1. As a result, all elements of the observation matrix must be assigned to the $1, \ldots, (k-1)$-th blocks. Therefore, typical Bayesian inference methods, such as Markov chain Monte Carlo (MCMC), can naturally avoid handling an infinite number of blocks.

# 5 Application to relational data analysis

**Relational model -** The BBP-based relational model is applied to the input matrix $\boldsymbol{X} := (X_{i,j})_{N \times M}$. We assume that $\boldsymbol{X}$ consists of categorical elements, that is, $X_{i,j} \in \{1, 2, \dots, H\}$, where $H \in \mathbb{N}$. The generative model can be constructed as follows. The BBP consists of i.i.d. uniform random variables $\boldsymbol{U} := (U_1, U_2 \dots)$ on $[0, 1]$, and i.i.d. beta random variables $\boldsymbol{\beta} := (\beta_1, \beta_2, \dots)$:

$$U_k \sim \text{Uniform}([0,1]), \quad \beta_k \sim \text{Beta}(1, \alpha) \quad (k = 1, 2, \dots), \tag{6}$$

where $\alpha$ is a non-negative hyper-parameter. For notational convenience, we also use $\boldsymbol{U}_k = (U_1, U_2 \dots, U_k)$ and $\boldsymbol{\beta}_k = (\beta_1, \beta_2, \dots, \beta_k)$. They correspond to a sample of rectangular partitioning on $[0, 1] \times [0, 1]$. The $k$-th block has a latent Dirichlet random variable $\boldsymbol{\phi}_k$:

$$\boldsymbol{\phi}_k \sim \text{Dirichlet}(\boldsymbol{\alpha}_0) \quad (k = 1, 2, \dots), \tag{7}$$

where $\boldsymbol{\alpha}_0 = (\alpha_0, \dots, \alpha_0)$ is a $H$-dimensional non-negative hyper-parameter. According to the AHK representation [6, 29, 32], each row and column of the input matrix is mapped into $[0, 1]$:

$$U_i^{\text{row}} \sim \text{Uniform}([0,1]) \ (i = 1, 2, \dots, N), \quad U_j^{\text{column}} \sim \text{Uniform}([0,1]) \ (j = 1, 2, \dots, M). \tag{8}$$

Finally, given the row locations $\boldsymbol{U}^{\text{row}} := (U_1^{\text{row}}, \dots, U_N^{\text{row}})$, the column locations $\boldsymbol{U}^{\text{column}} := (U_1^{\text{column}}, \dots, U_M^{\text{column}})$, the BBP parameters consisting of $\boldsymbol{U} = (U_1, U_2 \dots)$ and $\boldsymbol{\beta} = (\beta_1, \beta_2, \dots)$, and $(\boldsymbol{\phi}_1, \boldsymbol{\phi}_2, \dots)$, each element $X_{i,j}$ of the input matrix is drawn from the $H$-dimensional categorical distribution:

$$X_{i,j} \mid \boldsymbol{U}^{\text{row}}, \boldsymbol{U}^{\text{column}}, \boldsymbol{U}, \boldsymbol{\beta}, \boldsymbol{\phi}_{k(i,j)} \quad \sim \quad \text{Categorical}(\boldsymbol{\phi}_{k(i,j)}), \tag{9}$$

where $k(i, j)$ indicates the block index to which the point $(U_i^{\text{column}}, U_j^{\text{column}})$ belongs.

We compare the BBP-based relational model with the BNP stochastic block models based on RP: (1) The IRM [33]: the intermediate random function of the AHK representation is drawn from the product of the SBPs, and the concentration parameter is drawn from the $\text{Gamma}(1, 1)$ prior, as in [23]. (2) The MP [49]: the intermediate random function of the AHK representation is drawn from the MP, the budget parameter of which is set to 3, as in [23]. (3) The RTP [42]: we combine the product of the SBPs (also used in the aforementioned IRM) and the RTP is combined to construct the AHK representation.

**Bayesian inference -** For all models, we used an MCMC method that iterates over (1) drawing $\boldsymbol{U}^{\text{row}}$ and $\boldsymbol{U}^{\text{column}}$ (i.e., the corresponding locations on $[0, 1]$ of the rows and columns of the input matrix for the AHK representation), (2) updates of the current intermediate random function of the AHK representation (i.e., the current RP in the MCMC iterations), and (3) changing the complexity of the RP based on reversible jump schemes. To change the RP complexity of the MP and the RTP, we employ the methods in [53] and [42], respectively. For the reversible jump proposal of the BBP, a new block can be added, or the block with the largest label can be removed in the evolution of the BBP. The full description of our Bayesian inference method is provided in the supplementary material. The source code is available at `https://github.com/nttcslab/baxter-permutation-process`.

**Datasets -** We synthetically generated three relational matrices, with ground-truth partitions corresponding to **regular grid**, **hierarchical**, and **arbitrary** RP samples, respectively. Each matrix consists of $300 \times 300$ binary elements drawn from the beta-Bernoulli likelihood model. We also used four social network datasets [54, 35] (corresponding to Figure 1):

- **Wiki** (top-left) [1], consisting of 7115 nodes and 103689 edges with diameter 7.
- **Facebook** (top-right) [2], consisting of 4039 nodes and 88234 edges with diameter 8.
- **Twitter** (bottom-left) [3], consisting of 81306 nodes and 1768149 edges with diameter 7.
- **Epinion** (bottom-right) [4], consisting of 75879 nodes and 508837 edges with diameter 14.

For each data, we selected the top 1000 active nodes based on their interactions with others; subsequently we randomly sampled $500 \times 500$ matrix to construct the relational data, as in [23]. For model comparison, we held out $20\%$ cells of the input data for testing, and each model was trained by the MCMC using the remaining $80\%$ of the cells. We evaluated the models using perplexity as a criterion: $\text{perp}(\hat{X}) = \exp(-(\log p(\hat{X}))/N)$, where $N$ is the number of non-missing cells in the partitioned matrix $\hat{X}$.

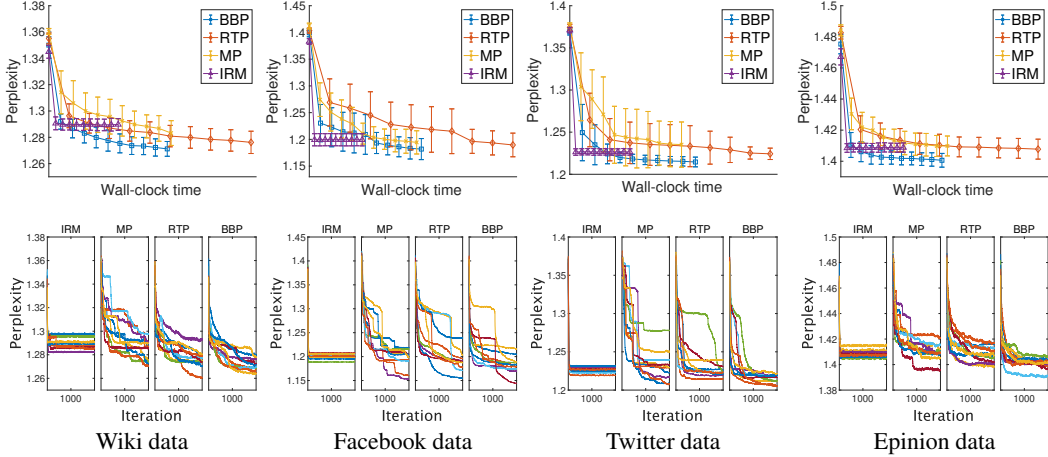

Figure 8: Experimental results of perplexity comparison. Each column corresponds to each real world data (The results for synthetic data are reported in the supplementary material). **Top:** Relationship between test perplexity (mean±std) evolution and wall-clock time. **Bottom:** Relationship between test perplexity evolution and 2000 MCMC iterations for 10 trials.

**Experimental results -** Table 1 and Figure 8 summarize the test perplexity comparison results. We recall that the *arbitrary* RP (covered by the BBP and the RTP) includes the *hierarchical* RP (corresponding to the MP) and the *regular grid* RP (corresponding to the IRM). Therefore, we can expect that the BBP (and the RTP) essentially does not degrade the predictive performance for any ground-truth partitions. However, in practice, there may be certain issues related to Bayesian inference, such as local optima and slow mixing. Fortunately, as shown in Table 1, the BBP exhibits better (at least comparable) performance than the other three models. It can also be seen that the RTP achieves a predictive performance similar to that of the BBP (Figure 8, bottom). However, as shown in Figure 8 (top), the RTP has high computational cost. We also observe that the IRM performs faster mixing of the MCMC iterations than the BBP. This implies that the BBP may be improved by using more sophisticated inference methods, including sequential Monte Carlo methods [34, 26], particle Markov chain Monte Carlo samplers [23, 8], and Bayesian combinatorial optimization methods [36, 11]; this is a further research direction.

Table 1: Perplexity comparison for real-world relational data analysis (mean±std)

|  | IRM [33] | MP [49] | RTP [42] | **BBP** (proposed) |
|---|---|---|---|---|
| Synth (regular grid) | **1.1791** ±0.0031 | 1.3690 ±0.0951 | 1.2709 ±0.0820 | 1.2136 ±0.0292 |
| Synth (hierarchical) | 1.2163 ±0.0145 | 1.2956 ±0.0913 | 1.2262 ±0.0314 | **1.2014** ±0.0105 |
| Synth (arbitrary) | 1.1299 ±0.0070 | 1.1983 ±0.0711 | 1.1406 ±0.0271 | **1.1161** ±0.0151 |
| Wiki | 1.2898 ±0.0045 | 1.2838 ±0.0094 | 1.2762 ±0.0085 | **1.2712** ±0.0056 |
| Facebook | 1.2012 ±0.0058 | 1.1944 ±0.0217 | 1.1895 ±0.0221 | **1.1818** ±0.0197 |
| Twitter | 1.2265 ±0.0038 | 1.2316 ±0.0209 | 1.2243 ±0.0067 | **1.2146** ±0.0058 |
| Epinion | 1.4088 ±0.0030 | 1.4098 ±0.0064 | 1.4078 ±0.0064 | **1.4006** ±0.0044 |

# 6 Conclusion

This paper has proposed new stochastic processes. Our main contributions are as follows: (1) We have presented the BNP model of the BP as a Markov process consisting of a sequence of i.i.d. uniform random variables on $[0, 1]$. Owing to the one-to-one correspondence between BP and FP, the model can also be used as a probabilistic model on the set of all possible FPs. (2) We combined the BPP with the BBP to obtain a stochastic process for arbitrary RPs. As in conventional methods, we applied this process to the AHK representation to construct a BNP stochastic block model for relational data, and compared its predictive performance with that of the IRM, MP, and RTP.

## Broader Impact

Clustering is one of the most fundamental machine learning tools for data analysis. The block-breaking process (BBP) can be regarded as a multi-dimensional extension of clustering and it has a potential to give a new perspective to relational data analysis, for it would reveal latent structures in relational data (or network data) in much more flexible manner than other existing clustering methods, without tuning the model complexity.

In fact, the BBP can extract latent clusters of relational data through rectangular partitioning (RP). While conventional models can express only limited classes of all possible RPs, the BBP can potentially capture arbitrary rectangular partitioning, keeping the central advantage of the Bayesian nonparametric (BNP) machine learning, and the BBP does not have to tune the model complexity regardless of the size of the input data. Therefore, the BBP will have a wide range of potential applications, including market research, pattern recognition, image processing, pre- and post- processing of data, and structure learning of network models. For example, the BBP can be combined with deep neural network (DNN) models as a prior on the network, which simultaneously learns the DNN parameters and the network structure. It may also be used to expose and identify biases in data. The source code of the BBP-based relational model is available at `https://github.com/nttcslab/baxter-permutation-process`, with which you can try and examine the BBP-based relational data analysis by yourself.

Our work is not facilitating any unethical aspects of machine learning technologies, by genuinely pursuing the development of Bayesian methods in many applications settings. However, as is often the case with any clustering methods (or more generally any predictive algorithms), our proposal can be misused in a variety of context. Since the BPP may reveal hidden clusters from any input relational matrices, unethical applications may lead to unexpected results due to unexpected cues. This problem is highly dependent on the choice of input data. Therefore, what is suitable as input data needs to be carefully considered from an ethical perspective.

## Funding disclosure

Funding in direct support of this work is from NTT Corporation, without any third party funding.

## Footnotes

[1] Plainly, these conditions are fundamental assumptions for dealing with *infinity*, that is, for BNP models to analyze finite subsets of potentially infinite data via infinite-dimensional probabilistic models.

[2]Let $\sigma_1 \ldots \sigma_n$ be a permutation on $\{1, \ldots, n\}$. We call $\sigma_i$ a *left-to-right maximum* if $\sigma_i > \sigma_j$ for all $j < i$. Similarly, we call $\sigma_i$ a *right-to-left maximum* if $\sigma_i > \sigma_j$ for all $j > i$.

[3]In this area of research, *projective* or *self-similar* RP is a very popular notion. Therefore, one might think that this projectivity property of the BPP is carried over into self-similarity of the corresponding FP (Fig.5, right). However, it is not true. For example, the FP corresponding to **3127564** is not self-similar to that of $Q_{7,6}\mathbf{3127564} = \mathbf{312564}$. The projectivity property of the BPP is entirely considered in the Baxter permutation domain, whose main purpose is the existence of a model of BPs on $\{1, \ldots, \infty\}$.

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
