[Supplementary Material]

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

$$U_k \sim \mathrm{Uniform}([0, 1]), \quad \beta_k \sim \mathrm{Beta}(1, \alpha) \quad (k = 1, 2, \ldots), \tag{6}$$

where $\alpha$ is a non-negative hyper-parameter. For notational convenience, we also use $\boldsymbol{U}_k = (U_1, U_2 \ldots, U_k)$ and $\boldsymbol{\beta}_k = (\beta_1, \beta_2, \ldots, \beta_k)$. They correspond to a sample of rectangular partitioning on $[0, 1] \times [0, 1]$. The $k$-th block has a latent Dirichlet random variable $\boldsymbol{\phi}_k$:

$$\boldsymbol{\phi}_k \sim \mathrm{Dirichlet}(\boldsymbol{\alpha}_0) \quad (k = 1, 2, \ldots), \tag{7}$$

where $\boldsymbol{\alpha}_0 = (\alpha_0, \ldots, \alpha_0)$ is a $H$-dimensional non-negative hyper-parameter. According to the AHK representation [6, 29, 32], each row and column of the input matrix is mapped into $[0, 1]$:

$$U_i^{\mathrm{row}} \sim \mathrm{Uniform}([0, 1]) \ (i = 1, 2, \ldots, N), \quad U_j^{\mathrm{column}} \sim \mathrm{Uniform}([0, 1]) \ (j = 1, 2, \ldots, M). \tag{8}$$

Finally, given the row locations $\boldsymbol{U}^{\mathrm{row}} := (U_1^{\mathrm{row}}, \ldots, U_N^{\mathrm{row}})$, the column locations $\boldsymbol{U}^{\mathrm{column}} := (U_1^{\mathrm{column}}, \ldots, U_M^{\mathrm{

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

## A Proofs

### A.1 Proof of Theorem 3.1

**Sketch -** We begin with a high level sketch. For any $n \in \mathbb{N}$ and $z_n = (\sigma_1 \ldots \sigma_n) \in \mathcal{Z}_n$, we consider to evaluate a lower bound of $\mu_n(z_n)$. We recall that, by construction, $z_n$ corresponds to i.i.d. uniform random variables $U_1, \ldots, U_n$ on $[0, 1]^n$. Here, it is not easy to calculate the probability of the series of $U_1, \ldots, U_n$ that corresponds to $z_n$. Therefore we introduce a special subset of $[0, 1]^n$, and let $A_{z_n}$ be the set that the order of $U_1, \ldots, U_n$ is consistent with $\sigma_1 \ldots \sigma_n$. We would like to emphasize that, for the BP sample $z_n = (\sigma_1 \ldots \sigma_n)$, the order of the corresponding latent variables $U_1, \ldots, U_n$ is not necessarily consistent with $\sigma_1 \ldots \sigma_n$. However, $A_{z_n}$ is certainly a subset whose corresponding BP sample is $z_n$, and it is fortunately easy to calculate probability that $U_1, \ldots, U_n$ belong to $A_{z_n}$. As a result, we can obtain a lower bound of $\mu_n(z_n)$.

**Full proof -** For any $n \in \mathbb{N}$ and $z_n = (\sigma_1 \ldots \sigma_n) \in \mathcal{Z}_n$, we explicitly evaluate a lower bound of $\mu_n(z_n)$. We introduce the projector $Q_{m,n} : \mathcal{Z}_m \to \mathcal{Z}_n$ $(n < m \in \mathbb{N})$, defined in Theorem 3.2: For a BP on $\{1, \ldots, m\}$, the projector $Q_{m,n}$ removes the largest $(m - n)$ labels of the permutation and generates a new BP on $\{1, \ldots, n\}$. We recall that, by construction, $z_n$ is derived from i.i.d. uniform random variables $U_1, \ldots, U_n$ on $[0, 1]^n$. First, we clarify the necessary and sufficient condition that $U_1, \ldots, U_n$ corresponds to $z_n$. Without loss of generality, we can suppose that the projection $Q_{n,m} z_n$ from $z_n$ to the BP on $\{1, \ldots, m\}$ $(m \le n)$ has left-to-right maxima $x_1^{(m)} < \cdots < x_{i_m}^{(m)} = m$ and right-to-left maxima $m = y_{j_m}^{(m)} > \cdots > y_1^{(m)}$. Then the series of $U_1, \ldots, U_n$ corresponds to $z_n$, if and only if the following holds:

$$U_{x_1^{(m)}} < \cdots < U_{x_{i_m}^{(m)}} = U_m = U_{y_{j_m}^{(m)}} < \cdots < U_{y_1^{(m)}} \quad \text{for all } m = 1, \ldots, n. \quad (10)$$

Next we introduce sufficiently small real $\epsilon > 0$ (more specifically, we set $0 < \epsilon < 1/(n+1)$), and consider the following subset $A_{z_n}$ of $[0, 1]^n$:

$$A_{z_n} = \left\{ U_1, \ldots, U_n \in [0, 1]^n \;\middle|\; \frac{i}{n+1} \le U_{\sigma_i} < \frac{i}{n+1} + \epsilon \quad (i = 1, \ldots, n) \right\}. \quad (11)$$

We can easily check that, if $U_1, \ldots, U_n \in A_{z_n}$, then the series of $U_1, \ldots, U_m$ satisfies Equation (10) for all $m = 1, \ldots, n$. This means that $A_{z_n}$ is the subset of $[0, 1]^n$ whose corresponding BP sample is equivalent to $z_n$. Therefore, we have

$$
\begin{aligned}
\mu_n(z_n) \;&>\; \int \cdots \int_{[0,1]^n} \mathbb{I}\left[ (U_1, \ldots, U_n) \in A_{z_n} \right] dU_1 \ldots dU_n \\
&=\; \int \cdots \int_{[0,1]^n} \prod_{i=1}^{n} \mathbb{I}\left[ \frac{i}{n+1} \le U_{\sigma_i} < \frac{i}{n+1} + \epsilon \right] dU_1 \ldots dU_n = \epsilon^n > 0. \quad (12)
\end{aligned}
$$

We complete the proof.

### A.2 Proof of Theorem 3.2

**Sketch -** It is sufficient to show that, for any $n \in \mathbb{N}$ and any BP sample $z_n \in \mathcal{Z}_n$ on $\{1, \ldots, n\}$, we have $\mu_n(z_n) = \mu_{n+1}(Q_{n+1,n}^{-1} z_n)$. Without loss of generality, we can suppose that $z_n$ has left-to-right maxima $x_1 < \cdots < x_i = n$ and right-to-left maxima $n = y_j > \cdots > y_1$. By construction, $z_n$ is derived i.i.d. uniform random variables $U_1, \ldots, U_n$ on $[0, 1]^n$. Moreover, as discussed in the main body (Section 3), we can assume

$$U_{x_1} < U_{x_2} < \cdots < U_{x_i} = U_n = U_{y_j} < U_{y_{j-1}} < \cdots < U_{y_1}. \quad (13)$$

Then, $\mu_n(z_n)$ can be intuitively expressed as follows:

$$\mu_n(z_n) = \int \cdots \int_{[0,1]^n} \mathbb{I}\left[ (U_1, \ldots, U_n) \text{ corresponds to } z_n \right] dU_1 \ldots dU_n. \quad (14)$$

Later, for full proof, we will introduce a function $f_{z_n}^{(n)}(U_1, \ldots, U_n)$, and simply write

$$\mu_n(z_n) = \int \cdots \int_{[0,1]^n} f_{z_n}^{(n)}(U_1, \ldots, U_n) dU_1 \ldots dU_n. \quad (15)$$

Figure 9: Illustration of function $f_{z_n}^{(m)}$ for case $i_m \leq i_{m-1}$. Note $U_m = U_{x_{i_m}^{(m)}} = U_{y_{j_m}^{(m)}}$. This figure shows that $U_m$ is drawn in the interval $[U_{x_{i_m-1}^{(m)}} = U_{x_{i_m-1}^{(m-1)}}, U_{x_{i_m}^{(m-1)}}]$.

On the other hand, the set $Q_{n+1,n}^{-1} z_n$ consists of a collection of BPs obtained by adding $n+1$ into one of admissible positions of $z_n$, which are the immediate left of $x_1 < \cdots < x_i = n$ and the immediate right of $n = y_j > \cdots > y_1$. Therefore, we have

$$
\mu_{n+1}(Q_{n+1,n}^{-1} z_n) = \int \cdots \int_{[0,1]^{n+1}} \left( \mathbb{I}[0 \leq U_{n+1} < U_{x_1}] + \cdots + \mathbb{I}[U_{x_{i-1}} \leq U_{n+1} < U_n] \right.
$$

$$
\left. + \mathbb{I}[U_n \leq U_{n+1} < U_{y_{j-1}}] + \cdots + \mathbb{I}[U_{y_1} \leq U_{n+1} < 1] \right) f_{z_n}^{(n)}(U_1, \ldots, U_n) dU_1 \ldots dU_{n+1}
$$

$$
= \int \cdots \int_{[0,1]^{n+1}} \mathbb{I}[0 \leq U_{n+1} < 1] f_{z_n}^{(n)}(U_1, \ldots, U_n) dU_1 \ldots dU_{n+1} = \mu_n(z_n). \quad (16)
$$

**Full proof -** Without loss of generality, it is sufficient to show that, for any $n \in \mathbb{N}$ and any BP sample $z_n \in \mathcal{Z}_n$ on $\{1, \ldots, n\}$, we have $\mu_n(z_n) = \mu_{n+1}(Q_{n+1,n}^{-1} z_n)$. By construction, $z_n$ is derived from i.i.d. uniform random variables $U_1, \ldots, U_n$ on $[0,1]^n$. We first recall the necessary and sufficient condition that the series of $U_1, \ldots, U_n$ corresponds to $z_n$. Without loss of generality, we can suppose that the projection $Q_{n,m} z_n$ from $z_n$ to the BP on $\{1, \ldots, m\}$ ($m \leq n$) has left-to-right maxima $x_1^{(m)} < \cdots < x_{i_m}^{(m)} = m$ and right-to-left maxima $m = y_{j_m}^{(m)} > \cdots > y_1^{(m)}$. Then $U_1, \ldots, U_n$ corresponds to $z_n$, if and only if the following holds:

$$
U_{x_1^{(m)}} < \cdots < U_{x_{i_m}^{(m)}} = U_m = U_{y_{j_m}^{(m)}} < \cdots < U_{y_1^{(m)}} \quad \text{for all } m = 1, \ldots, n. \quad (17)
$$

Next, for the BP sample $z_n$, we introduce a set of functions $f_{z_n}^{(m)} : [0,1]^m \to \{0,1\}$ ($m = 1 \ldots, n$), recursively defined as follows (see also Figure 9 and an example below):

$$
f_{z_n}^{(m)}(U_1, \ldots, U_m)
$$
$$
= \begin{cases} \mathbb{I}\left[U_{x_{i_m-1}^{(m)}} = U_{x_{i_m-1}^{(m-1)}} \leq U_m < U_{x_{i_m}^{(m-1)}}\right] f_{z_n}^{(m-1)}(U_1, \ldots, U_{m-1}) & (i_m \leq i_{m-1}) \\ \mathbb{I}\left[U_{y_{j_m}^{(m-1)}} \leq U_m < U_{y_{j_m-1}^{(m)}} = U_{y_{j_m-1}^{(m-1)}}\right] f_{z_n}^{(m-1)}(U_1, \ldots, U_{m-1}) & (\text{otherwise}) \end{cases} \quad (18)
$$

For example, we consider $z_9 = \mathbf{934128576}$ and $m = 8$. The projection $Q_{9,(m-1)} z_9 = Q_{9,7} z_9 = \mathbf{3412576}$ has

$$
x_1^{(7)} = \mathbf{3} < x_2^{(7)} = \mathbf{4} < x_3^{(7)} = \mathbf{5} < x_4^{(7)} = \mathbf{7}, \quad y_2^{(7)} = \mathbf{7} > y_1^{(7)} = \mathbf{6}. \quad (19)
$$

The projection $Q_{9,m} z_9 = Q_{9,8} z_9 = \mathbf{34128576}$ has

$$
x_1^{(8)} = \mathbf{3} < x_2^{(8)} = \mathbf{4} < x_3^{(8)} = \mathbf{8}, \quad y_3^{(8)} = \mathbf{8} > y_2^{(8)} = \mathbf{7} > y_1^{(8)} = \mathbf{6}. \quad (20)
$$

Given $f_{z_9}^{(7)}(U_1, \ldots, U_7)$, we can obtain $f_{z_9}^{(8)}(U_1, \ldots, U_8)$ as follows:

$$
f_{z_9}^{(8)}(U_1, \ldots, U_8) = \mathbb{I}[U_4 \leq U_8 < U_5] f_{z_9}^{(7)}(U_1, \ldots, U_7)
$$

$$
= \mathbb{I}\left[U_{x_2^{(8)}} = U_{x_2^{(7)}} \leq U_8 < U_{x_3^{(7)}}\right] f_{z_9}^{(7)}(U_1, \ldots, U_7). \quad (21)
$$

We recall that Equation (18) involves the term $\mathbb{I}\left[U_{x_{i_m-1}^{(m)}} = U_{x_{i_m-1}^{(m-1)}} \leq U_m < U_{x_{i_m}^{(m-1)}}\right]$, which corresponds to $i_m = 3$, and

$$
U_{x_{i_m-1}^{(m)}} = U_{x_{3-1}^{(8)}}, \qquad U_{x_{i_m-1}^{(m-1)}} = U_{x_{3-1}^{(8-1)}}, \qquad U_{x_{i_m}^{(m-1)}} = U_{x_3^{(8-1)}}. \quad (22)
$$

As stated above, we can obtain $f_{z_n}^{(n)}(U_1, \ldots, U_n)$. Then, it follows from the necessary and sufficient condition (Equation (17)) that $U_1, \ldots, U_n$ corresponds to $z_n$ that we have

$$\mu_n(z_n) = \int \cdots \int_{[0,1]^n} f_{z_n}^{(n)}(U_1, \ldots, U_n) dU_1 \ldots dU_n. \tag{23}$$

According to Corollary 2.3 (in the main body), the set $Q_{n+1,n}^{-1} z_n$ consists of a collection of BPs obtained by adding $n + 1$ into one of admissible positions of $z_n$, which are the immediate left of $x_1^{(n)} < \cdots < x_{i_n^{(n)}}^{(n)} = n$ and the immediate right of $n = y_{j_n}^{(n)} > \cdots > y_1^{(n)}$. Therefore, we have

$$
\begin{aligned}
\mu_{n+1}(Q_{n+1,n}^{-1} z_n) = & \int \cdots \int_{[0,1]^{n+1}} \mathbb{I}\left[0 \le U_{n+1} < U_{x_1^{(n)}}\right] f_{z_n}^{(n)}(U_1, \ldots, U_n) dU_1 \ldots dU_{n+1} \\
& + \cdots + \int \cdots \int_{[0,1]^{n+1}} \mathbb{I}\left[U_{x_{i_n-1}^{(n)}} \le U_{n+1} < U_n\right] f_{z_n}^{(n)}(U_1, \ldots, U_n) dU_1 \ldots dU_{n+1} \\
& + \int \cdots \int_{[0,1]^{n+1}} \mathbb{I}\left[U_n \le U_{n+1} < U_{y_{j_n-1}^{(n)}}\right] f_{z_n}^{(n)}(U_1, \ldots, U_n) dU_1 \ldots dU_{n+1} \\
& + \cdots + \int \cdots \int_{[0,1]^{n+1}} \mathbb{I}\left[U_{y_1^{(n)}} \le U_{n+1} < 1\right] f_{z_n}^{(n)}(U_1, \ldots, U_n) dU_1 \ldots dU_{n+1} \\
= & \int \cdots \int_{[0,1]^{n+1}} \mathbb{I}\left[0 \le U_{n+1} < 1\right] f_{z_n}^{(n)}(U_1, \ldots, U_n) dU_1 \ldots dU_{n+1} \\
= & \int \cdots \int_{[0,1]^n} f_{z_n}^{(n)}(U_1, \ldots, U_n) dU_1 \ldots dU_n = \mu_n(z_n). \quad (24)
\end{aligned}
$$

We complete the proof.

## B  Details of Bayesian Inference

### B.1  Joint probability density

The BBP-based relational model involves the row locations $\boldsymbol{U}^{\text{row}} = (U_1^{\text{row}}, \ldots, U_N^{\text{row}})$, the column locations $\boldsymbol{U}^{\text{column}} = (U_1^{\text{column}}, \ldots, U_M^{\text{column}})$, the BBP parameters consisting of $\boldsymbol{U} = (U_1, U_2 \ldots)$ and $\boldsymbol{\beta} = (\beta_1, \beta_2, \ldots)$. The joint probability density function (joint PDF) is factorized to the following form:

$$
p\left(\boldsymbol{X}, \boldsymbol{U}^{\text{row}}, \boldsymbol{U}^{\text{column}}, \boldsymbol{U}, \boldsymbol{\beta}\right) = \left(\prod_{n=1}^{N} p_{\text{uniform}}\left(U_n^{\text{row}}\right)\right) \left(\prod_{m=1}^{M} p_{\text{uniform}}\left(U_m^{\text{column}}\right)\right)
$$
$$
\times \left(\prod_{k=1}^{\infty} p_{\text{uniform}}\left(U_k\right)\right) \left(\prod_{k=1}^{\infty} p_{\text{beta}}\left(\beta_k\right)\right) p_{\text{obs.}}(\boldsymbol{X} \mid \boldsymbol{U}^{\text{row}}, \boldsymbol{U}^{\text{column}}, \boldsymbol{U}, \boldsymbol{\beta}), \tag{25}
$$

where $p_{\text{uniform}}$ and $p_{\text{beta}}$ indicate the uniform PDF and the beta PDF, respectively, and

$$
p_{\text{obs.}}(\boldsymbol{X} \mid \boldsymbol{U}^{\text{row}}, \boldsymbol{U}^{\text{column}}, \boldsymbol{U}, \boldsymbol{\beta}) \propto \prod_{k=1}^{\infty} \left(\frac{\Gamma(H\alpha_0)}{\Gamma(H\alpha_0 + \sum_{h=1}^{H} \mathcal{N}_{k,h})} \prod_{h=1}^{H} \frac{\Gamma(\alpha_0 + \mathcal{N}_{k,h})}{\Gamma(\alpha_0)}\right), \tag{26}
$$

where $\mathcal{N}_{k,h}$ denotes the number of elements in both the $k$-th block and the $h$-th category of the categorical distribution.

It is not easy to directly deal with the above joint PDF due to an infinite number of products, and therefore it is not straightforward to obtain the simulation of the posterior distribution of the parameters. However, there exists a variety of tractable inference methods, including

- **Finite truncation -** Sufficiently large natural number is chosen in advance. It is used for a bound of the model dimensions.
- **Finite but unlimited model [?] -** As proposed in the context of the Dirichlet process mixture, the Poisson distribution is employed as the prior for the model dimensions.

- **Slice sampling [54] or retrospective sampler [47]** - Infinite number of parameters can be artificially avoided by some kind of adaptive threshold.
- **Reversible jump Markov chain Monte Carlo (RJMCMC) method [55]** - Simulation of the posterior distribution on spaces of varying model dimensions is allowed.

Specifically, in the following, we describe an RJMCMC method, which can be straightforwardly applied to the others with slight modification.

### B.2 Reversible jump Markov chain Monte Carlo

To obtain an RJMCMC algorithm, we reformulate the joint PDF as a mixture of varying model dimensions. As is discussed in the main body (Section 4), for a finite input matrix, there exists a natural number $k^* < \infty$ such that each elements of the input matrix is assigned into either of the $1 \ldots, k^*$-th block. Therefore, the joint PDF (Equation (25)) can be reformulated as follows:

$$p\left(\boldsymbol{X}, \boldsymbol{U}^{\mathrm{row}}, \boldsymbol{U}^{\mathrm{column}}, \boldsymbol{U}, \boldsymbol{\beta}\right) = \sum_{k^*=1}^{\infty} \mathbb{P}\left[\sum_{h=1}^{H} \mathcal{N}_{k^*,h} > 0 \wedge \sum_{k=k^*+1}^{\infty} \sum_{h=1}^{H} \mathcal{N}_{k,h} = 0\right]$$

$$\times \left(\prod_{n=1}^{N} p_{\mathrm{uniform}}\left(U_n^{\mathrm{row}}\right)\right)\left(\prod_{m=1}^{M} p_{\mathrm{uniform}}\left(U_m^{\mathrm{column}}\right)\right)\left(\prod_{k=1}^{k^*+1} p_{\mathrm{uniform}}\left(U_k\right)\right)\left(\prod_{k=1}^{k^*-1} p_{\mathrm{beta}}\left(\beta_k\right)\right)$$

$$\times p_{\mathrm{obs.}}\left(\boldsymbol{X} \mid \boldsymbol{U}^{\mathrm{row}}, \boldsymbol{U}^{\mathrm{column}}, \boldsymbol{U}, \boldsymbol{\beta}\right). \quad (27)$$

In the following, for simplicity, the finite subset of the model parameter is denoted by $\boldsymbol{\theta}_k := (\boldsymbol{U}^{\mathrm{row}}, \boldsymbol{U}^{\mathrm{column}}, \boldsymbol{U}_{k+1}, \boldsymbol{\beta}_{k-1})$. The first term of the right side indicates the case that the $k^*$-th block is just *active* and all $k > k^*$-th blocks are not active. Fortunately, we can explicitly evaluate it, when the input matrix $\boldsymbol{X}$, the row locations $\boldsymbol{U}^{\mathrm{row}}$, the column locations $\boldsymbol{U}^{\mathrm{column}}$, and the finite subsets of the BBP parameters $\boldsymbol{U}_{k^*+1}$ and $\boldsymbol{\beta}_{k^*-1}$ are given. For simplicity, we abbreviate its conditional probability to $p_{\mathrm{comp.}}(k^* \mid \boldsymbol{X}, \boldsymbol{\theta}_{k^*})$.

Now we will evaluate $p_{\mathrm{comp.}}(k^* \mid \boldsymbol{X}, \boldsymbol{\theta}_{k^*})$, which is the conditional probability that $k^*$-th block is just *active* and all $k > k^*$-th blocks are not active, given $\boldsymbol{X}$ and $\boldsymbol{\theta}_{k^*}$. We recall some notations. Let $U_{\mathrm{max}}^{\mathrm{row}}$ and $U_{\mathrm{max}}^{\mathrm{column}}$ be $\max\{U_i^{\mathrm{row}} \mid i = 1, \ldots, N\}$ and $\max\{U_j^{\mathrm{column}} \mid j = 1, \ldots, M\}$, respectively. We additionally let $z_k^{\mathrm{row}}$ and $z_k^{\mathrm{column}}$ be the vertical and horizontal locations of the top-left corner of the $k$-th block, respectively. As is discussed in the main body (Section 4), there exists a natural number $k < \infty$ such that $U_{\mathrm{max}}^{\mathrm{row}} < z_{k+1}^{\mathrm{row}}$ and $U_{\mathrm{max}}^{\mathrm{column}} < z_{k+1}^{\mathrm{column}}$ hold. Moreover, $C_{k^*+1}$ indicates the set of the blocks adjacent to the **left** (or **top**) of the $(k^* + 1)$-th block, and $C_{k^*+1}^{\mathrm{min}}$ is a block in $C_{k^*+1}$ with the minimum **width** (or **height**) $l_{k^*}$. For example, we consider the case that $C_{k^*+1}^{\mathrm{min}}$ has the minimum width $l_n$, and the minimum block index in $C_{k^*+1}$ is $c_{k^*+1}$. Then we have

$$\mathbb{P}\left[\sum_{k=k^*+1}^{\infty} \sum_{h=1}^{H} \mathcal{N}_{k,h} = 0\right] = \int_0^1 \mathbb{I}\left[U_{\mathrm{max}}^{\mathrm{column}} < (1-\beta)z_{C_{k^*+1}^{\mathrm{min}}}^{\mathrm{column}} + \beta\right]$$

$$\times \mathbb{I}\left[U_{\mathrm{max}}^{\mathrm{row}} < z_{c_{k^*+1}}\right] p_{\mathrm{beta}}(\beta)\, d\beta \quad (28)$$

Using the cumulative distribution function (CDF) $I_{\mathrm{beta}}(\beta; 1, \alpha)$ (i.e., the incomplete beta function) of the beta random variable $\beta \sim \mathrm{Beta}(1, \alpha)$, we have

$$\mathbb{P}\left[\sum_{k=k^*+1}^{\infty} \sum_{h=1}^{H} \mathcal{N}_{k,h} = 0\right] = \mathbb{I}\left[U_{\mathrm{max}}^{\mathrm{row}} < z_{c_{k^*+1}}\right]\left(1 - I_{\mathrm{beta}}\left(\frac{U_{\mathrm{max}}^{\mathrm{column}} - z_{C_{k^*+1}^{\mathrm{min}}}^{\mathrm{column}}}{1 - z_{C_{k^*+1}^{\mathrm{min}}}^{\mathrm{column}}}; 1, \alpha\right)\right). \quad (29)$$

Then, we obtain

$$p_{\mathrm{comp.}}(k^* \mid \boldsymbol{X}, \boldsymbol{\theta}_{k^*}) = \mathbb{I}\left[\sum_{h=1}^{H} \mathcal{N}_{k^*,h} > 0\right] \mathbb{I}\left[U_{\mathrm{max}}^{\mathrm{row}} < z_{c_{k^*+1}}\right]$$

$$\times \left(1 - I_{\mathrm{beta}}\left(\frac{U_{\mathrm{max}}^{\mathrm{column}} - z_{C_{k^*+1}^{\mathrm{min}}}^{\mathrm{column}}}{1 - z_{C_{k^*+1}^{\mathrm{min}}}^{\mathrm{column}}}; 1, \alpha\right)\right) \quad (30)$$

Fortunately, given $\boldsymbol{X}$ and $\boldsymbol{\theta}_{k^*}$, all terms of the right side can be explicitly calculated. Finally, we obtain the mixture of all finite models, which is suitable to the RJMCMC method:

$$p\left(\boldsymbol{X},\boldsymbol{\theta}\right) = \sum_{k^*=1}^{\infty} p\left(\boldsymbol{X},\boldsymbol{\theta}_{k^*}, k^*\right) = \sum_{k^*=1}^{\infty} p_{\text{comp.}}(k^* \mid \boldsymbol{X},\boldsymbol{\theta}_{k^*}) p_{\text{model}}(\boldsymbol{\theta}_{k^*}) p_{\text{obs.}}(\boldsymbol{X} \mid \boldsymbol{\theta}_{k^*}), \quad (31)$$

where $p_{\text{obs.}}(\boldsymbol{X} \mid \boldsymbol{\theta}_{k^*})$ is Equation (26), and $p_{\text{model}}(\boldsymbol{\theta}_k)$

$$= \left(\prod_{n=1}^{N} p_{\text{uniform}}\left(U_n^{\text{row}}\right)\right) \left(\prod_{m=1}^{M} p_{\text{uniform}}\left(U_m^{\text{column}}\right)\right) \left(\prod_{k=1}^{k^*+1} p_{\text{uniform}}\left(U_k\right)\right) \left(\prod_{k=1}^{k^*-1} p_{\text{beta}}\left(\beta_k\right)\right).$$

The RJMCMC method involve the Metropolis-Hastings (MH) type algorithm that move a simulation analysis between models defined by $(\boldsymbol{\theta_k}, k)$ to $(\boldsymbol{\theta_{k'}}, k')$ with different dimensions $k$ and $k'$. We begin with a high level sketch of the general RJMCMC. In the following, one of the most fundamental version of the RJMCMC is described. If the current state of the Markov chain is $(\boldsymbol{\theta_k}, k)$, then the transition to a new state is as follows.

*Step 1* Propose a visit to a new model complexity $k'$ with a *proposal* probability $R(k \rightarrow k')$.

*Step 2* Sample an auxiliary random variable $\boldsymbol{v}$ from a *proposal* density $q(\boldsymbol{v} \mid \boldsymbol{\theta}_k, k, k')$.

*Step 3* Set $(\boldsymbol{\theta}_{k'}, \boldsymbol{v}') = g_{k,k'}(\boldsymbol{\theta}_k, \boldsymbol{v})$, where $g_{k,k'}$ is a bijection between $(\boldsymbol{\theta}_k, \boldsymbol{v})$ and $(\boldsymbol{\theta}_{k'}, \boldsymbol{v}')$, where $\boldsymbol{v}$ and $\boldsymbol{v}'$ play the role of matching the dimensions of the model parameters such that $(\boldsymbol{\theta}'_k, \boldsymbol{v}')$ has the same dimension as $(\boldsymbol{\theta}_k, \boldsymbol{v})$.

*Step 4* The acceptance probability of the new model $(\boldsymbol{\theta}_{k'}, k')$ can be calculated as

$$\min\left(1, \frac{p\left(\boldsymbol{X},\boldsymbol{\theta}_{k'}, k'\right)}{p\left(\boldsymbol{X},\boldsymbol{\theta}_k, k\right)} \frac{R(k' \rightarrow k) q(\boldsymbol{v}' \mid \boldsymbol{\theta}_{k'}, k', k)}{R(k \rightarrow k') q(\boldsymbol{v} \mid \boldsymbol{\theta}_k, k, k')} \left|\frac{\partial g_{k,k'}(\boldsymbol{\theta}_k, \boldsymbol{v})}{\partial(\boldsymbol{\theta}_k, \boldsymbol{v})}\right|\right) \quad (32)$$

Based on this framework, we can obtain a specific MCMC sampler, that iterates over the following three sub-routines:

**Update model complexity -** We can employ a simple random walk on the Markov process corresponding to the BPP as the proposal $R(k \rightarrow k')$ for new model complexity, that is, (1) adding a new block or (2) removing the current bottom-right block. For the proposal $q(\boldsymbol{v} \mid \boldsymbol{\theta}_k, k, k')$, we can choose the simplest sampler called *independent sampler* of the RJMCMC framework:

(1) **Adding a new block, i.e.,, the case that $k' = k + 1$** - We first draw $\boldsymbol{v} = (v_1, v_2)$ as $v_1 \sim \text{Uniform}([0, 1])$ and $v_2 \sim \text{Beta}(1, \alpha)$. Then the function $g_{k,k'}(\boldsymbol{\theta}_k, \boldsymbol{v})$ regards $v_1$ and $v_2$ as $U_{k+2}$ and $\beta_k$, respectively, and generates $\boldsymbol{\theta}_{k'} = \boldsymbol{\theta}_{k+1}$.

(2) **Removing the current bottom-right block, i.e., the case that $k' = k-1$** - We first remove $U_{k+1}$ and $\beta_{k-1}$ from $\boldsymbol{\theta}_k$. Then the function $g_{k,k'}(\boldsymbol{\theta}_k, \boldsymbol{v})$ regards the resulting $\boldsymbol{\theta}_{k-1}$ as $\boldsymbol{\theta}_{k'}$.

Finally, the acceptance/rejection scheme (Equation (32)) is applied.

**Update row and column locations -** We can easily obtain Gibbs sampling on the row and column locations $\boldsymbol{U}^{\text{row}}$ and $\boldsymbol{U}^{\text{column}}$, similar to [55], since their posterior distributions are piece-wise constant. However, this Gibbs sampler may require high computational cost for sufficiently large input matrix. In such cases, we also have another option. As in [23], we can also use the prior (i.e., uniform distribution on $[0, 1]$) as the proposal, and apply the MH acceptance/rejection scheme.

**Update rectangular partitioning -** For the fixed model complexity $k$, the current rectangular partitioning consisting $(U_1, \ldots, U_{k+1})$ and $(\beta_1, \ldots, \beta_{k-1})$ can be updates based on the MH algorithm. We draw a new candidate of each of $(U_1, \ldots, U_{k+1})$ and $(\beta_1, \ldots, \beta_{k-1})$ from its prior, and apply the MH acceptance/rejection scheme.