[Reviews · NeurIPS 2020]

Review 1

Summary and Contributions: The paper presents a novel Bayesian nonparametric clustering model based on a class of permutations known as Baxter permutations, showing how this approach makes it possible to capture floorplan partitioning, which go beyond classical product partitions and tree-based partitions. The paper also presents a generalization of stick-breaking to a "block-breaking process", and shows how the latter adds an appropriate sizing vector that allows floorplans to be turned into general rectangular partitions.

Strengths: This is a creative paper that adds significantly to the repertoire of combinatorial stochastic processes, by making use of some interesting combinatorics, and showing how these processes connect to a class of flexible rectangular partitioning processes. This approach connects naturally to MCMC-based inference. The experimental results are strong, with meaningful comparisons to nontrivial baselines.

Weaknesses: Not really a weakness, but I look forward to seeing a longer paper that develops characterizations of the BPP and BBP stochastic processes.

Correctness: Yes.

Clarity: Yes.

Relation to Prior Work: Yes.

Reproducibility: Yes

Additional Feedback:


Review 2

Summary and Contributions: This paper aims to propose a new stochastic space-partitioning process to generate arbitrary rectangular partitions. It first finds the one-to-one correspondence between the Baxter permutation and the floorplan partition, and makes use of some nice properties of the Baxter permutation to construct and enumerate floorplan partitions. Equipped with these properties, the authors propose a construction strategy to generate Baxter permutations from Z_n to Z_{n+1} by using a novel sampling method and prove the projectivity of the proposed Baxter permutation process (BPP). As the generated floorplan has no block sizes, the authors further propose a block-breaking process (BBP) to assign the size to each block in the floorplan partition generated by a BPP, resulting arbitrary rectangular partitions. The proposed BBP is applied as a graphon in modeling multi-arrays on some relational modeling benchmark datasets. Some empirical tests show that the proposed method outperforms IRM [32], MP [48], and RTP [41]. >>>>>>>>Update after authors' response<<<<<<<< My question about the evolution of a floorplan partition (FP) according to the Baxter permutation (BP) process has been addressed. I'm happy to see a new 'flavor' added to the stochastic space-partitioning process family.

Strengths: 1) A stochastic space-partitioning process being able to partition the space into arbitrary rectangular partitions is an interesting and challenging research topic. 2) This paper finds a new angle to construct a stochastic space-partitioning process different from the existing relevant works, such as IRM, MP, BSP, RBP, RTP, etc. -- This is exciting. 3) Making use of the correspondence between the Baxter permutation and the floorplan partition to generate a new block on an existing floorplan partition is interesting and novel. It is nice to see the Baxter permutation process is projective. 4) Extending the discrete and size-less Bartex permutation process (BPP) to the block breaking process (BBP) on a continuous space is interesting. BPP shares the same interesting properties as the stick-breaking process and shows potentials for further development. 5) The graphic illustrations are helpful in understanding the ideas.

Weaknesses: 1) The projectivity of the Baxter permutation process (BPP) is *not* carried over to its corresponding floorplan partition (footnote 3 on page 6). As the projectivity of the block-breaking process (BBP) is built on BPP, BBP is also not projective. 2) Descriptions of some important steps are missing, such as 1) how to insert a new block in an existing floorplan partition (lack of explanations for Figure 5(right)); 2) how to expand the domain of a floorplan partition from an n*n 2-array to an (n+1)*(n+1) 2-array (lack of explanation for Figure 6(right)), and see more minor missing details in “Clarity” below. 3) The authors have provided a comprehensive collection of relevant works; however, the proposed BBP is only compared to three of them. I understand that the authors might think that they may only need to compare with axis-aligned partitioning methods, but if the proposed BBP could be compared with methods with sloped cuts to see which type of flexibility is more powerful in real-world relational modeling, it would be more interesting.

Correctness: To the best of my understanding, I do not see visible technical issues.

Clarity: The paper is well organized and presented. The idea is quite easy to follow. I am particularly grateful to the graphic illustrations in Figures 2—7, which is very helpful to help understand concepts and algorithms, given that some parts of texts are not very readable. Some examples as below: 1) It would be great if the authors can elaborate more on the concept, definition, and the reason of avoidance of vincular patterns in Line 69. 2) In Lines 82 and 84, “shifting … until … and pulling the attached T-junctions” is difficult to follow, thanks to the accompanied figures to help understand. 3) For the evolution of a floorplan partition according to the BPP: - How to insert a new block in an existing floorplan partition is not introduced for Figure 5(right); - How to expand the domain of a floorplan partition from an n*n 2-array to an (n+1)*(n+1) 2-array is not introduced for Figure 6(right).

Relation to Prior Work: This paper has done a comprehensive review on the prior work. It is obvious that the proposed approach follows a very different idea from the existing relevant methods.

Reproducibility: Yes

Additional Feedback: - Please provide the reference of Algorithm 2 in Section 2.2.2. - How to select the parameters for the beta distributions in BBP?


Review 3

Summary and Contributions: The Baxter Permutation Process provides a generalization of the Mondrian process so that an arbitrary tiling with hyperrectables is supported. In particular, the Mondrian process (MP) is a Bayesian nonparametric version of a decision tree. Since MP is defined by a tree, each cut must extend to the edges of the hyperrectangle being cut. This means that the first cut must extend from -infinity to +infinity, and the second cut (should it be perpendicular) must extend from infinity to the level of the first cut (or, if it's parallel, again from -infinity to +infinity). This extensive cut nature is not incredibly terrible, but could lead to a lack of local modelling. The proposed Baxter process would allow cuts to stop before they extend totally along an axis, due to the specific generative process provided by the authors (section 2.3). As such, this work may be a great addition to Bayesian nonparametric literature. There are three hurdles that must be surpased in order to recommend this work: 1. Distinction from the Regular Tiling Process (Nakano et al. 2014 JMLR). This is previous work that also describes an arbitrary tessellation. Reading the author's paper, it's not clear what the difference is. The authors reference this work, saying "(3) The RTP [41]: we combine the product of the SBPs (also used in the aforementioned IRM) and the RTP is combined to construct the AHK representation." It's not clear to me what this means, even after reviewing acronyms. What is the difference between the regular tilling process and the author's proposed model? 2. Proof of projectivity. Bayesian nonparametric models must be consistent, or projective. This means that if we subset the space upon which the process is defined, we get a conditional version of the original model (e.g., Kolmogorov's extension theorem). This is required for correct inference, and also to decouple the method from a fixed sample size. "The first property is that BPs are closed under removing the largest label, leading to the projectivity property of the BPP for Kolmogorov's extenstion theorem ..." The authors appear to well establish projectivity. 3. Reasonable performance on real datasets. The authors demonstrate this in Table 1. What are the settings for the models considered? Does each model have a limit on the number of blocks in the partition, or are they run until each partition has only a single label in the training set? What are the sizes of these datasets? I'd also like to mention a potential application to approximate nearest neighbours. This is a new area in machine learning, based on doing a faster and more geometric job on KNN. Work towards this would be supported by the sort of local tiles that the Baxter permutation process produces. Overall: I believe this is a great paper, but I do need to know more about the differentiation from RTP [41] before recommending.

Strengths: Claims are sound. Significance is great. Novelty is unclear: relation to RTP [41] must be more explored. Relevance to NeurIPS community must be known. The authors provide great responses to my questions, I'm raising my score accordingly.

Weaknesses: For real data experiments, can we know more about the properties of the datasets, for example sample sizes? Comparisons with more models could be done.

Correctness: The claims and methods are of good quality and seem correct.

Clarity: The paper is well written.

Relation to Prior Work: This paper solves an important problem in Bayesian nonparametrics. Rectangular Tiling Process [41] reference also solves the same problem - what are the main differences? Why does the author's method outperform RTP in their real data experiments?

Reproducibility: Yes

Additional Feedback: Thank you.


Review 4

Summary and Contributions: This paper propose a new nonparametric Bayesian tool which is based on the Baxter permutation. The stochastic process based on this Baxter permutation is named as the Baxter permutation process (BPP), which can be used in modeling the relational data. According to the results presented in this paper, this new BPP can work in a easy way and more close to the Bayesian inference.

Strengths: This paper provides a novel tool leveraging a new stochastic process that not used in the nonparametric Bayesian analysis before, the Baxter process. It discuss this BPP with both theoretical and numeric details. Also it compares with existed tools, such as the floorplan partitioning and the stick-breaking process. This new tool provides a significant alternative to existed nonparametric Bayesian tools.

Weaknesses: In Sec. 4 the BPP is discussed in details. While as a discrete time Markov chain, the BPP is not discussed with its time complexity or other quantitative analysis of computation expense (or I didn't find it). Would the authors please talk more on this point?

Correctness: The new process as explained in the paper makes sense and the numerical results look convincing.

Clarity: The paper is pretty well written with good illustrative figures.

Relation to Prior Work: In Sec. 2.2.1 and Sec. 2.2.2, the proposed BPP is compared with the FP in details.

Reproducibility: Yes

Additional Feedback:

[Author Response · NeurIPS 2020]

We thank reviewers for their thoughtful and positive feedback. **For significance**, we are encouraged that all reviewers (R1, R2, R3, R4) agree significance of our work, and also be glad that R3 recognizes insightful potential applications. **For novelty**, we are pleased that reviewers find that our paper is *creative* (R1), *finds a new angle* (R2) of model construction based on *a very different idea from the existing relevant methods* (R2), and *provides a novel tool (...) that not used (...) before* (R4). We will answer R3's concern about the difference between our model and the conventional rectangular tiling process (ICML2014) below. **For clarity**, we are glad that reviewers find that our paper is well written (R1,R3), *the idea is quite easy to follow* (R2), and *the new process makes sense* (R4). We will answer R2's main concern about how to obtain the evolution of the underlying floorplan partitioning according to our Baxter permutation process. **We address some specific comments below and will incorporate all the feedbacks in the revised paper.**

For R1 - We again appreciate your positive feedback. We agree with R1 in the sense that combinatorial stochastic processes involving permutations must be one of the promising directions of BNP for further studies and developments.

@R2 - **How to obtain the evolution of a floorplan partition (FP) according to the Baxter permutation (BP) process**: Instead of direct transformations from a FP with $n$ blocks to a FP with $n+1$ blocks, the evolution of a FP is obtained only through the underlying evolution of a BP by using Algorithm 2 (mapping from BPs into FPs). For example, we consider an evolution of a BP from **312564** to **3127564** (Figure 5, right). We apply Algorithm 2 to both **312564** and **3127564**, and obtain the corresponding FPs to **312564** and **3127564**, respectively. Similarly, for Figure 6 (right), Algorithm 2 is applied to all possible BPs independently.

@R2 - "*the proposed BBP could be compared with methods with sloped cuts to see which type of flexibility is more powerful in real-world relational modeling*": We would like to consider a variant of sloped (oblique) cuts so that the proposed BBP can be extended to a model with the sloped cuts, analogous to the relationship between the binary space partitioning-tree process [19, 25] and the Mondrian process [50]. We agree that recent developments of BNP models with sloped cuts are a very interesting and promising direction of research, and expect that our BPP could also contribute to this issue. We hope that such extension of the BBP could cover the partitions which cannot be easily expressed by the previous models, e.g., pin-wheel structure with sloped cuts shown in the right figure.

@R3 - **Difference between our model (BBP) and the rectangular tiling process (RTP)**: The BPP grows along with the number of blocks, while the conventional RTP grows along with the size of virtual grids (right figure). The RTP constructs a probabilistic generative model that directly generates rectangular partitioning (RP) of grids (i.e., a matrix) with infinite size. As a result, the RTP has too complicated procedures for the model construction, and is not well-suited for Bayesian inference. Specifically, it is very difficult to establish the *effective* Metropolis-Hastings (MH) algorithm for the update steps of the current RP sample drawn from the RTP, while the BPP (and the BBP) can naturally lead to such steps. Consequently, the RTP tends to have low acceptance ratio for such MH steps, which is the most important difference and the practical advantage of the BPP over the RTP.

@R3 - **For experiments,** "*Does each model have a limit on the number of blocks in the partition, or do they run until each partition has only a single label in the training set?*": No, each model is allowed to have unlimited number of blocks without finite truncations. Since Bayesian nonparametric models practically behave like *parsimonious* models, infinite number of meaningless blocks (containing no observation data) are ignored, and *active* blocks must be always finite for finite observation data. For our MCMC inference, the number of active blocks is allowed to be variable order.

@R3 - **For experiments,** "*Can we know more about the properties of the datasets, for example sample sizes?*": As in Lines 230–234, for each real-world data, we employed the *active* 500-by-500 binary matrix extracted from the original very large and *sparse* matrix. For example, **Wiki** data (Wikipedia vote network) [1] consists of 7115 nodes and 103689 edges with diameter 7. We will add this kind of detailed explanations for all data.

@R4 - "*While as a discrete time Markov chain, the BPP is not discussed with its time complexity or other quantitative analysis of computation expense*": **From a viewpoint of generative models** - It takes $\mathcal{O}(K)$ for the BPP (BBP) to draw a FP (RP) with $(K+1)$ blocks given a FP (RP) sample with $K$ blocks. As a result, the evolution of the BPP works with reasonable computational cost, even when significantly many blocks are required. We also emphasize that the conventional Mondrian process (MP) [50] (which can be also expressed as a Markov process) has the same computational order for its evolution. **From a viewpoint of Bayesian inference** - Given an $M$-by-$N$ input observation matrix, the MCMC calculations plainly involves (a) $\mathcal{O}(K)$: random growth or regression of the Markov process (i.e., updates of the BBP), (b) $\mathcal{O}(M+N)$: random locations of rows and columns of the input matrix, and (c) $\mathcal{O}(KMN)$: data assignments of $MN$ elements to *active* $K$ blocks. (Note: $K$ is allowed to be variable order on each MCMC iteration.) This computational order is same as the MP, which we can also see in the experimental results (Figure 8, top).

[Meta-Review · NeurIPS 2020]

This article proposes a novel Bayesian nonparametric model for relational data. The model exploits a connection between rectangular partitioning models and Baxter permutations to introduce a new class of rectangular partitioning process, called Baxter permutation processes. As mentioned by the reviewers, this paper is creative and brings a new perspective on such class of models. The properties of the model are well described, and the proposed approach compared to suitable baselines.